# Physiological Effects and Mechanisms of *Chlorella vulgaris* as a Biostimulant on the Growth and Drought Tolerance of *Arabidopsis thaliana*

**DOI:** 10.3390/plants13213012

**Published:** 2024-10-28

**Authors:** Jinyoung Moon, Yun Ji Park, Yeong Bin Choi, To Quyen Truong, Phuong Kim Huynh, Yeon Bok Kim, Sang Min Kim

**Affiliations:** 1Smart Farm Research Center, Korea Institute of Science and Technology (KIST), Gangneung Institute of Natural Products, Gangneung 25451, Republic of Korea; jy77mun@kist.re.kr (J.M.); yunji0825@changwon.ac.kr (Y.J.P.); choiyeongbin@kist.re.kr (Y.B.C.); ttquyen_vn@kist.re.kr (T.Q.T.); 623024@kist.re.kr (P.K.H.); 2Gyeongnam Bio and Anti-Aging Core Facility Center, Changwon National University, Changwon 51140, Republic of Korea; 3Natural Product Applied Science, KIST School, University of Science and Technology, Seoul 02792, Republic of Korea; 4Department of Medicinal and Industrial Crops, Korea National University of Agriculture & Fisheries, Jeonju 54874, Republic of Korea; yeondarabok@korea.kr

**Keywords:** microalgae, *Chlorella vulgaris*, *Arabidopsis thaliana*, biostimulant, drought tolerance, cell-free supernatant, stomata aperture, auxin, IAA5, glucosinolate

## Abstract

Microalgae have demonstrated biostimulant potential owing to their ability to produce various plant growth-promoting substances, such as amino acids, phytohormones, polysaccharides, and vitamins. Most previous studies have primarily focused on the effects of microalgal biostimulants on plant growth. While biomass extracts are commonly used as biostimulants, research on the use of culture supernatant, a byproduct of microalgal culture, is scarce. In this study, we aimed to evaluate the potential of *Chlorella vulgaris* culture as a biostimulant and assess its effects on the growth and drought tolerance of *Arabidopsis thaliana*, addressing the gap in current knowledge. Our results demonstrated that the *Chlorella* cell-free supernatant (CFS) significantly enhanced root growth and shoot development in both seedlings and mature *Arabidopsis* plants, suggesting the presence of specific growth-promoting compounds in CFS. Notably, CFS appeared to improve drought tolerance in *Arabidopsis* plants by increasing glucosinolate biosynthesis, inducing stomatal closure, and reducing water loss. Gene expression analysis revealed considerable changes in the expression of drought-responsive genes, such as *IAA5*, which is involved in auxin signaling, as well as glucosinolate biosynthetic genes, including *WRKY63*, *MYB28*, and *MYB29*. Overall, *C. vulgaris* culture-derived CFS could serve as a biostimulant alternative to chemical products, enhancing plant growth and drought tolerance.

## 1. Introduction

Microalgae are unicellular photosynthetic microorganisms with diverse applications in biofuels, food supplements, and wastewater treatment [1,2,3]. They are valued for their use in dietary supplements and as feed additives in aquaculture and animal husbandry, as their biomass is rich in essential amino acids, carbohydrates, lipids, and vitamins [4,5]. Additionally, microalgae have been recognized for their potential use as biofertilizers to promote plant growth [6,7]. *Chlorella* species, which are unicellular green algae, can thrive in various environments, demonstrating remarkable adaptability and ecological importance [8]. In particular, *Chlorella vulgaris*, one of the most well-known green algae, has been extensively studied for its nutritional and pharmaceutical value. Furthermore, *C. vulgaris* is considered to be a “green biofuel factory” owing to its high lipid content, serving as a sustainable alternative to fossil fuels [9].

Various green algae, particularly *Chlorella* sp., have been extensively studied for their biostimulant properties [10,11]. For instance, *C. fusca* treatment can improve the growth of Chinese chives in field conditions [12]. Additionally, the application of live *Chlorella* cells using a soil-drenching method resulted in increased biomass of *Medicago truncatula* plants [13]. Other studies have demonstrated the beneficial effects of *C. vulgaris* and *C. pyrenoidosa* as biofertilizers on salt-affected soils, enhancing the growth of crops such as lettuce, rice, eggplant, and cucumber [14]. In this study, we chose *C. vulgaris* microalga for its broad application and large-scale production potential. The soil drenching method is commonly used for the application of biofertilizers and biostimulants. *Arabidopsis thaliana*, a model plant widely used in scientific research owing to its small size, short life cycle, rapid reproduction, and genetic tractability, shares physiological similarities with important crops such as rice, wheat, and tomato [15,16,17]. Previous studies have shown that various microalgal biofertilizers can influence the growth, development, and defense responses of *A. thaliana*. For instance, the application of *C. fusca* can increase the resistance of *A. thaliana* to *Pseudomonas syringae* infection [18]. In the culture supernatant, Ɒ-lactic acid released by *C. fusca* was identified as a defense-priming agent.

Drought is a major environmental stress factor that negatively impacts plant development and growth, causing biochemical, molecular, and physiological changes [19]. Plants may employ various adaptive strategies, such as hormonal regulation, reduced stomatal conductance, and increased antioxidant levels, to cope with water stress [20,21]. Microalgal biofertilizers have been shown to enhance drought tolerance in plants. For example, *C. vulgaris* application under drought conditions improved biochemical processes in broccoli, protecting it from oxidative damage [22]. Similarly, spraying wheat plants with Carolina algal extract mitigated the effects of drought stress, thereby enhancing growth and yield parameters [23]. Furthermore, it was reported that Sargassum and Carolina extracts were more effective than commercial algal treatments at increasing plant tolerance to drought. Another study demonstrated that *Chlorella* induced stomatal closure in the epidermal peel of *Vicia faba* in dose- and time-dependent manners [24]. The results of that study revealed that *Chlorella* treatment induced stomatal closure via NADPH oxidase-dependent reactive oxygen species (ROS) production in epidermal strips and improved the intrinsic water use efficiency in leaves.

Drought stress triggers molecular responses in plants, which are initiated by various growth substances or phytohormones, including ABA, auxin, brassinosteroids, cytokinin, ethylene, and strigolactones. In the brassinosteroid signaling pathway, the transcription factor BRI1-EMS-Suppressor 1 (BES1) acts as a negative regulator by inhibiting the drought-responsive gene *RD26* [25]. In the strigolactone signaling pathway, the repressor Suppressor of MAX2-Like6 (SMXL6) negatively regulates plant responses and adaptation to drought stress [26]. Auxin signaling factors IAA5, IAA6, and IAA19 inhibit the expression of transcription factor *WRKY63*, leading to the transcriptional activation of *MYB28* and *MYB29*, thereby increasing glucosinolate (GLS) levels and enhancing drought tolerance [27]. ABA-mediated induction of *WRKY18* and *WRKY40* expression leads to increased accumulation of protein products that form both homo- and heterocomplexes through physical interactions [28]. *WRKY18* and *WRKY60* positively regulate, while *WRKY40* negatively regulates, plant responses to ABA, probably by modulating ABA-regulated genes. ABREs are cis-acting promoter elements bound by AREBs/ABFs, members of the bZIP transcription factor (TF) family. Since ABA regulates most of its target genes through AREBs/ABFs, the ABRE element is recognized as a key signature for drought-responsive genes regulated by the ABA-dependent pathway. The dehydration-responsive element (DRE) (A/GCCGAC), present in various drought-responsive genes, is recognized by DREB2 proteins. DREB2A, which belongs to the AP2/ERF (Apetala2 and ethylene-responsive factors) TF family [29], is slightly induced by ABA but highly induced by dehydration, highlighting its role in the ABA-independent response to drought stress [30].

Despite recent advances, the specific mechanisms by which microalgae enhance plant growth and drought resistance remain poorly understood. Here, we aimed to elucidate the biological mechanisms underlying the effects of *C. vulgaris* treatments on the growth and drought tolerance of *A. thaliana*. The effects of different *Chlorella* treatments on the growth indices of seedlings and mature *Arabidopsis* plants were evaluated. Subsequently, drought resistance in plants was determined based on the changes in water content, stomatal aperture, and malondialdehyde (MDA) levels. The underlying mechanisms of drought resistance through the auxin signaling pathway and GLS biosynthesis were also investigated. Our findings suggest that *Chlorella* culture supernatant could be utilized as an alternative biostimulant to synthetic fertilizers, promoting growth and drought tolerance in agricultural applications.

## 2. Results

### 2.1. A. thaliana Growth Enhancement by C. vulgaris Cell-Free Supernatant (CFS)

In this study, *A thaliana* Columbia-0 (Col-0) plants were grown with various *Chlorella* culture supplements, including whole *Chlorella* culture suspension (CS), *Chlorella* biomass (CB), and *Chlorella* cell-free supernatant (CFS), also referred to as filtrate supernatant. The results obtained from the plants grown with different *Chlorella* supplements were normalized to those from the control group, which was supplied with water only. To assess the effects on seedling growth, *Arabidopsis* seedlings were grown for 14 days on Murashige and Skoog (MS) medium supplemented with 10% and 2% CS, CB, and CFS (Figure 1A). Notably, the seedlings grown on CFS-supplemented MS showed significant increases in both primary root length and cotyledon length when compared with those grown on CB- or CS-supplemented medium (Figure 1B,C, *p* < 0.001). In particular, the longest primary root length was just below 100 mm, and the longest cotyledon length nearly reached 10 mm. While cotyledons doubled in length with 10% CFS supplement, CB and CS supplements resulted in either no significant difference or only slight increases compared with those in the control group.

Subsequent experiments were conducted to assess the effects of *Chlorella* treatments on the growth and development of mature *Arabidopsis* plants. Fourteen-day-old seedlings cultivated on untreated MS medium were transplanted into soil treated with various concentrations of water-diluted CB, CFS, and CS, applied every three days. The effects of *Chlorella* supplements on growth indices such as rosette size, leaf length, plant height, silique length, and thickness were evaluated. Two weeks after transplantation, shoot diameter and young leaf length increased with CFS diluted ten-fold (10×) and twofold (2×) with water (Figure 2A,E,F). Notably, treatment with 2× CFS supplement resulted in *Arabidopsis* plants with larger rosette sizes (~30 mm in diameter) and longer young leaves (20 mm long). By contrast, neither CB nor CS treatments significantly promoted young leaf growth.

Four weeks after transplantation, *Arabidopsis* plants had matured and developed siliques. Consistent improvements in plant height, mature leaf length, and silique morphology were observed in *Arabidopsis* plants grown in CFS-supplemented soil (Figure 2B–D,G,H). The plants treated with 10× CFS, 2× CFS, and 2× CS exhibited significantly longer mature leaves (*p* < 0.05). Additionally, silique length and thickness were significantly enhanced by CFS supplements (*p* < 0.001). The effects of CFS were concentration-dependent at the early stage, whereas both 10× and 2× CFS supplements exerted similar effects on leaf growth and silique formation.

### 2.2. Drought Tolerance Improvement in A. thaliana by C. vulgaris CFS

To investigate the effects of *Chlorella* supplements on drought tolerance, *Arabidopsis* seedlings were grown on 10% CB-, CFS-, and CS-supplemented MS medium. After two weeks, the seedlings were transferred to Petri dishes covered with Parafilm and incubated for 90 min under continuous illumination to induce water deficit. Significant morphological changes observed in the control group demonstrated rapid stress induction within a short period (Figure 3A). Similarly, the fresh biomass of plants in both the control and *Chlorella* supplement-treated groups was reduced by more than 50%, reflecting water loss (Figure 3B). Notably, the reduction in fresh biomass varied among the *Chlorella* treatments, with the lowest biomass loss occurring in seedlings grown on 10% CFS-supplemented medium. Seedlings in the control group experienced the highest biomass loss due to drought stress.

To further assess recovery from drought stress, 14-day-old seedlings were subjected to water stress and then regrown on MS medium supplemented with CB, CFS, or CS for 24 h. The effects of *Chlorella* supplements on the recovery ability of the *Arabidopsis* seedlings were evaluated. No signs of recovery were observed in both the control and treatment groups after water stress (Figure 3C). However, the seedlings grown on CFS-supplemented medium exhibited a slight increase in fresh weight, which was significantly higher than that in the other groups (Figure 3D).

The effects of *Chlorella* supplements on drought tolerance were also observed in mature *Arabidopsis* plants. Fourteen-day-old seedlings grown on untreated MS medium were transplanted into soil; 10× CB, CFS, and CS solutions were supplied every three days for two weeks before halting irrigation for 11 days. The severe impact of water deficit on plant growth was evident in the control and CS-treated plants (Figure 4A). Notably, there were no significant morphological changes in the control group even after resuming irrigation for four days. Consistently, the lowest biomass was observed in the control plants, which was only 25% of the initial plant biomass (Figure 4B). Among the *Chlorella* treatments, the CS-treated plants exhibited a 50% biomass loss after drought stress. Notably, the CB- and CFS-treated plants showed a slight increase in growth despite water limitation.

### 2.3. C. vulgaris CFS Regulates Several Physiological Effects in Response to Drought Stress

#### 2.3.1. Water Loss, Stomatal Aperture, and MDA Accumulation in *A. thaliana*

Subsequently, experiments were conducted to investigate the potential mechanisms underlying the impact of *Chlorella* supplements on the enhanced resistance of mature plants to water stress. After withholding the water supply, the fresh weight was continuously recorded at intervals of 0, 15, 30, 45, 60, 120, and 240 min to determine the extent of water loss. The results indicated that the water content consistently decreased, but this reduction varied among the control and treatment groups. In particular, the CFS-treated plants exhibited significantly slower water loss than those treated with CB or CS (Figure 5A, *p* < 0.001). CS treatment resulted in a water loss pattern similar to that in the control group.

Furthermore, the CFS-treated plants exhibited reduced stomatal apertures when compared with the plants subjected to other treatments, indicating that CFS promoted stomatal closure (Figure 5B,C). In addition, malondialdehyde (MDA) production was examined in the leaves of two-week-old *Arabidopsis* plants treated with CB, CFS, or CS, as well as in those subjected to drought stress for 11 days. MDA, a chemical compound generated as a secondary metabolite, is an indicator of lipid peroxidation and cell membrane collapse induced by ROS during stress. The results showed a marked decrease in MDA content in *Arabidopsis* plants treated with CFS after drought stress compared with those before drought stress (Figure 5D), suggesting enhanced drought tolerance mediated by CFS.

#### 2.3.2. Gene Expression Analysis

Various plant hormones, including ABA, auxins, brassinosteroids, and strigolactones, play crucial roles in stress responses [25,26,27,28,29,30]. The expression levels of key genes involved in auxin signaling, GLS biosynthesis, and stress responses, including *BES1, RD26*, *SMXL6*, *IAA5*, *IAA6*, *IAA19*, *WRKY63*, *MYB28*, *MYB29*, *WRKY18*, *WRKY40*, *WRKY60*, *AREB1*, *ABF3*, and *DREB2A*, were investigated in the leaves of four-week-old *Arabidopsis* plants grown in soil treated with 10× CB, CFS, and CS.

The results showed that the investigated genes exhibited variations in expression levels across different *Chlorella* treatments (Figure 6).

Notably, most of the genes investigated were upregulated by *Chlorella* supplements under drought conditions. In particular, *IAA5* was exclusively upregulated over 20-fold, *IAA6* by over five-fold, and *MYB28* and *MYB29* by over four-fold in the leaves of plants treated with CFS (Figure 6). By contrast, the *WRKY40*, *WRKY18*, *WRKY63*, and *RD26* genes were highly downregulated after *Chlorella* treatments. Notably, *WRKY63* was exclusively downregulated two-fold in the leaves of plants treated with CFS.

GLSs play essential roles in plant defense and innate immunity. It has been suggested that GLS accumulation may induce stomatal closure, thereby improving drought tolerance [27]. Accordingly, we determined the total GLS content of *Arabidopsis* plants grown with CB, CFS, and CS supplements to assess their potential role in the drought-responsive system. The highest GLS content (~20 µmol g^−1^) was observed in plants treated with CFS; contrastingly, neither CS nor CB treatment improved the total GLS content in *Arabidopsis* plants (Figure 7).

## 3. Discussion

Among the numerous benefits of microalgal cultivation, previous studies have explored the potential applications of microalgal biostimulants in supporting sustainable agriculture [31]. In particular, previous results demonstrated biostimulating and biofertilizing properties of various microalgal species, including *C. vulgaris* and other microalgae. For example, sugar beet seeds treated with *C. vulgaris* extract exhibited superior germination performance compared to those treated with *Scenedesmus quadricauda* extract [32]. Similarly, the application of *C. vulgaris* extracts improved cucumber growth compared to extracts from other *Chlorella* sp. and *Chlamydopodium fusiforme* [33]. Another study examined the biostimulating effects of extracts from *C. vulgaris*, *Nannochloropsis salina*, and *Spirulina platensis* on common bean plants [34]. The results demonstrated that these extracts effectively enhanced bean growth parameters and nutritional value. While biostimulant effects considerably vary based on extract concentration, host plants, and microalgal species, most microalgae exhibit comparable effects.

A well-known example is that *Spirulina*-based biostimulants positively influence seed germination, improve growth, and enhance stress tolerance. Application of *Spirulina* extracts at different concentrations significantly increased pepper yield and quality parameters, such as fruit number, weight, and chlorophyll content, compared to the control treatment [35]. Notably, the exocellular polysaccharide (ESP) recovered from the spent culture medium of *Spirulina* sp. also exhibited growth-promoting effects, leading to increased biomass productivity [36]. Compared to the untreated control (water only), application of the EPS solution significantly enhanced root volume and shoot height in basil and spinach plants. Similarly, supplementation with the EPS recovered from *D. salina* culture supernatant increased tomato root and shoot biomass and mitigated the negative impacts of salinity stress on plant growth [37].

In microalgal cultivation, CFS, a byproduct, is often wasted after biomass collection and overlooked in research. Our research highlights the biostimulant potential of *Chlorella*-derived CFS, which promoted the growth of both seedlings and mature *Arabidopsis* plants, resulting in increased cotyledon and root length, enhanced leaf development, and improved silique hardening. These findings align with previous research reporting the positive effects of both CS and CFS on the growth of wheat plants [38]. Similarly, Alling et al. reported a significant increase in the germination index in tomatoes and barley when treated with CFS obtained from *Chlorella* cultured in municipal wastewater [39].

In this study, we also observed the dose-dependent impact of *Chlorella* treatments, which was consistent with prior findings. Specifically, Minaoui et al. (2024) found that a 25% concentration of CFS improved wheat germination, whereas a 50% concentration exhibited inhibitory effects [38]. While CFS showed growth-promoting effects on *Arabidopsis* plants, previous research investigating the effects of CB, CS, and CFS treatments on the growth of “Red Russian” kale demonstrated that CFS inhibited plant growth but stimulated the accumulation of bioactive compounds [40].

These findings suggest that CFS has the potential to serve as a biostimulant; however, various factors can affect the consistent effectiveness of CFS-based biostimulants. In the present study, 10% CFS supplementation exhibited the most significant impact on *Arabiodopsis* plants; contrastingly, a 25% CFS solution was the optimal treatment for promoting wheat growth [38]. Although *Chlorella* treatments showed effects comparable to those of microalgal species, it is essential to conduct further research across different plant models under laboratory and field conditions. Specifically, maize, rice, and wheat are economically important crops that substantially contribute to the global food supply.

Environmental constraints such as water stress can substantially hinder plant growth and development. Previous studies have indicated that microalgal treatments can enhance the plant defense system under stress conditions. For example, *C. vulgaris* biomass extracts have been shown to reduce the negative effect of drought stress on broccoli plants, improving biomass production and biochemical composition [22]. Similarly, treatment with *C. sorokiniana* biomass promoted the root development of maize plants and significantly alleviated the impacts of water deficit and nitrogen deficiency [41].

While a few studies have demonstrated the potential growth-promoting effects of filtrate supernatant, the impact of CFS under stress conditions has remained unclear [38]. Therefore, the present study provides preliminary evidence that CFS application might mitigate the inhibitory effects of water constraints on plant growth. Specifically, our results showed that 10% CFS supplement significantly decelerated water loss caused by drought stress. This may be due to the induction of stomatal closure, which helps to retain water in the leaves. Furthermore, oxidative stress levels were significantly reduced by CFS treatment, suggesting the activation of specific mechanisms in response to water deficit. In line with the previous results of Kusvuran S. (2021), CB treatment showed a slight positive effect on stress alleviation [22].

However, CS, referring to the whole culture suspension, did not exhibit beneficial effects on either plant growth or stress resistance. This suggests that rapid centrifugation may lead to the secretion of plant growth-promoting substances by live *Chlorella* cells into the supernatant. Additionally, the sterilization of CB, CFS, or CS before application could account for the differing effects of CS and CFS. Thermal treatment, specifically autoclaving, was performed to minimize contamination affecting *Arabidopsis* cultivation. *Chlorella* cells are characterized by the presence of a thick cell wall, which can be disrupted by various methods, including mechanical, physical, thermal, and chemical techniques. Autoclaving at high temperatures and pressure can result in the destruction of the rigid cell wall, leading to the release of nutrient compounds such as lipids and carbohydrates into the surrounding medium [42,43]. Furthermore, *Chlorella* biomass is rich not only in macronutrients but also in microelements such as Na, K, Ca, P, Mg, and Fe [44]. These mineral elements are also present in BG-11 culture medium, which implies that the total content of these elements in CS is higher than that in CFS samples [40]. Excessive amounts of specific minerals could inhibit plant growth. For example, copper (Cu) may affect the availability of other minerals and hinder nutrient uptake in plants [45].

The relationship between GLS biosynthesis and stomatal closure, regarding drought tolerance, has been reported [46]. GLSs are the most abundant secondary metabolites primarily found in the family Brassicaceae. Upon cell disruption, the myrosinase enzyme hydrolyzes GLSs, producing various hydrolysis products, such as isothiocyanates and nitriles. Under drought conditions, GLS accumulation or the formation of isothiocyanates may directly or indirectly regulate stomatal closure [47,48,49]. Notably, a recent study reported that treatment with varying concentrations of live *Chlorella* cells induced stomatal closure [24]. The regulation of stomatal closure is critical for plants to minimize adverse effects and enhance survival under water stress. Thus, the ABA-dependent pathway has been extensively studied in stomatal regulation [50,51,52]. ABA accumulation may facilitate Ca^2+^ ion uptake into guard cells and activate anion and outward K^+^ channels. This reduces turgor pressure inside guard cells, inducing stomatal closure and minimizing water loss through transpiration.

Next, we attempted to unravel the underlying mechanisms of drought resistance attributed to *Chlorella* treatments, providing new insights into auxin signaling and GLS biosynthesis. Phytohormones, such as auxin and ABA, play important roles in stress regulation, serving as secondary messengers to mediate downstream responses [53]. The AUX/IAA gene family, which includes 29 genes identified in *A. thaliana*, plays a central role in auxin signaling under stress conditions such as drought, salinity, and heat stress [54].

Notably, our results showed that the expression of *IAA5* increased nearly 25-fold in the leaves of plants grown in CFS-supplemented soil compared to those grown with CB and CS supplements. Other AUX/IAA family genes, including *IAA6* and *IAA19*, exhibited lower expression levels but were slightly upregulated under *Chlorella* supplements. Regarding GLS biosynthesis, the expression of *WRKY63* specifically decreased by more than two-fold with CFS supplement, while the expression levels of *MYB28* and *MYB29* were significantly increased by CFS and CS supplements. As mentioned, the *IAA5*, *IAA6*, and *IAA19* genes are required for drought tolerance in *Arabidopsis* [27]. *IAA5-*, *IAA6-*, and *IAA19*-knockout mutants exhibited physiological changes, such as reduced GLS accumulation and defective stomatal regulation, resulting in drought susceptibility. The *WRKY63* transcription factor likely inhibits the expression of genes regulating aliphatic GLS biosynthesis [46].

Therefore, CFS supplement might specifically upregulate the *IAA5* gene, inhibiting the expression of *WRKY63* and promoting GLS biosynthesis. Only CFS induced the upregulation of the *IAA5* gene, suggesting that certain compounds present in CFS specifically interact with the promoter of *IAA5* to regulate its expression.

Previous findings indicated that the expression of AUX/IAA family members, such as *IAA5,* was increased by exogenous factors, including ABA, auxin, salicylic acid, PEG, and NaCl [55]. Interestingly, the production of phytohormones in microalgal cells has been detected. In particular, *C. sorokiniana* grown under phototrophic conditions may contain 0.93 µg∙g^−1^ auxins and 20.87 µg∙g^−1^ gibberellins [56]. Mazur et al. (2001) detected the presence of indole-3-acetic acid (IAA) in the culture media of the green algae *Scenedesmus armatus* and *C. pyrenoidosa* [57]. *C vulgaris* biomass contains high concentrations of phytohormones, including cytokinins, auxins (0.58  ±  0.03 µg∙mL^−1^), and abscisic acid [58]. Therefore, the upregulation of the investigated AUX/IAA genes in *Arabidopsis* leaves could be induced by the presence of exogenous auxin derived from *Chlorella* culture.

In summary, CFS treatment promoted the growth of *Arabidopsis* plants and improved drought tolerance, which was associated with physiological changes, including enhanced GLS production, increased stomatal closure, and reduced water loss (Figure 8). *IAA5*, a member of the AUX/IAA family, may serve as a key gene involved in regulating the transcriptional cascade leading to GLS biosynthesis. Specifically, *IAA5* overexpression suppressed *WRKY63*, resulting in the induction of transcription factors *MYB28* and *MYB29* involved in GLS biosynthesis.

The results of the present study support the potential of recycling the byproduct from *Chlorella* culture as a biostimulant. Notably, this study indicates the potential of CFS to improve drought tolerance while unraveling the underlying mechanism involving the auxin signaling pathway. However, this study has some limitations, including a lack of comparison with modified BG-11 medium and the selection of study samples. First, the variations in plant growth and drought resistance are likely the result of additive effects from the mixture of inorganic and organic substances present in *Chlorella* culture supernatant. In addition to trace BG-11 components, *C. vulgaris* cells can secrete various substances, while the disrupted cells generate numerous biochemical compounds. Unfortunately, this study could not distinguish the above-mentioned aspects due to the absence of a medium treatment (modified BG-11). However, a previous study reported the non-significant effect of medium treatment on plant growth [38]. Additionally, our straightforward results certainly support the recycling of CFS for further applications in crop production, which aligns with the biorefinery concept of microalgal cultivation and sustainable agriculture. In particular, *Chlorella* biomass can be collected for other applications, such as food supplements, biofuel, and animal feed, while the culture supernatant can be used in crop production. The discrepancy between CS and CFS could be due to the release of various compounds or byproducts when the cells are destroyed, potentially interfering with their function as a biostimulant. Given the limited information available, this phenomenon warrants further investigation in future studies. Second, using other economically important crops would enhance the significance of our research and the direct applicability of our findings. Given these limitations, in the future, experiments should be conducted on these plants under field conditions to validate the effectiveness of CFS as a potent biostimulant.

## 4. Materials and Methods

### 4.1. Cultivation and Preparation of C. vulgaris Treatments

*C. vulgaris* (AG20696) was obtained from the Korean Collection for Type Cultures (KCTC) at the Korea Research Institute of Bioscience and Biotechnology in Jeongeup, Republic of Korea. *C. vulgaris* was cultivated in 2 L glass bottles in a growth chamber at 24 ± 2 °C with a light intensity of 200 μmol·m^−2^·s^−1^ using BG-11 culture medium. Seven-day-old cultures (the end of the exponential phase) with a cell density of 1 × 10^7^ cells·mL^−1^ were used for treatment and the preparation of *Chlorella* stock solutions.

The treatments and their descriptions are presented in Table 1. Briefly, CS refers to the whole *Chlorella* culture, while CB and CFS refer to the biomass and supernatant fractions of the *Chlorella* culture, respectively. To collect these fractions, *Chlorella* cultures were centrifuged at 3500× *g* for 20 min at 4 °C. Subsequently, CS, CB, and CFS were diluted with either MS medium or water for further treatments on seedlings and mature plants, respectively. Each solution was added to MS medium at final concentrations of 2% and 10%. For treating mature plants, *Chlorella* samples were diluted 50-, 10-, or 2-fold with distilled water (50×, 10×, and 2×) and applied to the soil every three days.

### 4.2. Plant Materials and Growth Conditions

Wild-type *A. thaliana* Col-0 seeds were sterilized twice with 70% ethanol and rinsed four times with distilled water before being planted on 0.5× MS medium containing 1% (*w*/*v*) sucrose and 0.8% plant agar (Duchefa Biochemie, BH Haarlem, The Netherlands) in square Petri dishes (100 × 100 mm, SPL). First, seedlings were grown in MS medium or in CS-, CB-, and CFS-supplemented MS medium for 14 days. Second, seedlings that had grown sufficiently in untreated MS medium were later transplanted into soil mixed with 70% (*v*/*v*) cocopeat, 10% (*v*/*v*) peat moss, 8% (*v*/*v*) perlite, 7% (*v*/*v*) vermiculite, and 5% (*v*/*v*) zeolite. *Chlorella* supplements, including 50×, 10×, and 2×, were applied every three days. Only water was used in the control group for normalization. Plants were grown in a growth chamber under controlled conditions: a 16 h:8 h light/dark cycle, 21 ± 1 °C, and 50% humidity.

### 4.3. Phenotype Observation of Seedlings and Mature Plants of A. thaliana

The growth phenotypes of 14-day-old Col-0 seedlings treated with 50-fold and 10-fold dilutions of CB, CFS, and CS were observed, focusing on primary root length and cotyledon length. For soil-grown plants, treatments were applied every three days, and phenotypes such as rosette size and leaf length were measured. After 4 weeks of growth, plant height and silique size were assessed at both 2 and 4 weeks. The data represent the mean values of five replicates ± standard deviation (SD), with the measurements performed using ImageJ software version 1.53 (National Institutes of Health, Bethesda, MD, USA).

### 4.4. Drought Stress Treatment

Fourteen-day-old seedlings treated with 10-fold dilutions of CB, CFS, and CS were subjected to drought stress by transferring them to Petri dishes covered with Parafilm and incubating them for 90 min under light. The phenotypes and fresh weights were recorded before and after the drought treatment. Untreated seedlings were also subjected to drought stress and then immediately transplanted to MS medium treated with CB, CFS, or CS for recovery observations. The fresh weight was measured before drought stress, after stress, and after re-watering. The data represent the mean values of 15 replicates ± SD.

For soil-grown plants, 14-day-old seedlings were transplanted into pots containing soil and treated with CB, CFS, or CS diluted 10× with distilled water once every three days for two weeks before irrigation was stopped for 11 days. After re-watering was performed for four days, the fresh weight was measured. The data represent the mean values of four replicates ± SD.

### 4.5. Measurement of Stomata Aperture

The abaxial epidermis of fully expanded leaves from two-week-old plants was peeled, cut into strips, and placed in liquid MS medium. Stomata were digitized using a Nikon Eclipse TS100 microscope, and the pore width and length of the stomatal pores were measured from digital images using ImageJ. The stomatal aperture was calculated based on these measurements. The data represent 150 measurements ± SD from three plants.

### 4.6. Analysis of MDA Contents

Lipid peroxidation was assessed by determining the MDA level, which is indicative of oxidative stress in plant tissues. MDA levels were evaluated using the thiobarbituric acid (TBA) reaction method, as described by Heath and Packer (1968) [59]. Leaf samples were collected from two-week-old *Arabidopsis* plants both before and after being subjected to drought stress for 10 days. Each sample, weighing approximately 50 mg, was immediately frozen in liquid nitrogen upon harvest to preserve its biochemical integrity. The frozen samples were ground in a pre-chilled mortar and pestle with two volumes of ice-cold 0.1% (*w*/*v*) trichloroacetic acid (TCA). The homogenate was then centrifuged at 15,000× *g* for 15 min to obtain the supernatant.

For the MDA assay, 1 mL of the supernatant was added to 2 mL of TBA solution (0.5% *w*/*v* in 20% TCA *w*/*v*). The mixture was heated in a water bath at 95 °C for 30 min and then rapidly cooled in an ice bath to terminate the reaction. After cooling, the samples were centrifuged again at 10,000× *g* for 10 min at 4 °C. The absorbance of the supernatant was measured at 532 nm, and the non-specific absorbance at 600 nm was subtracted from the 532 nm reading to correct for any background interference. The MDA concentration was calculated using an extinction coefficient of 155 mM^−1^·cm^−1^. Each measurement was replicated three times, and the data are presented as the mean ± SD.

### 4.7. Measurement of Water Loss in A. thaliana Plants

To assess the impact of *Chlorella* treatments on water retention in *Arabidopsis*, we measured the water loss in *Arabidopsis* plants subjected to drought stress. Two-week-old soil-grown plants treated with CB, CFS, and CS, all diluted 10-fold, were used for this experiment. The aerial parts of the well-watered plants were detached and weighed immediately to record their initial fresh weights (0 min). The detached samples were then left to dry under ambient conditions, and their weights were recorded at specific time intervals: 15, 30, 45, 60, 90, 120, and 240 min. The water content was calculated as a percentage of the initial fresh weight. The experiment was replicated three times, and the data are presented as the mean value ± SD.

### 4.8. Total RNA Isolation and Quantitative RT-PCR

Total RNA was extracted from the leaves of well-grown *Arabidopsis* plants to analyze gene expression levels. The harvested leaves were placed in 1.5 mL Eppendorf tubes, frozen in liquid nitrogen, and ground into a fine powder using a cold pestle and mortar. Total RNA was isolated from the powdered tissue using an RNA preparation kit (Geneaid, New Taipei City, Taiwan) following the manufacturer’s instructions. The isolated RNA was then resuspended in 30 µL of RNase-free water. The concentration and purity of the RNA were assessed using a Nanodrop spectrophotometer, targeting a concentration of approximately 300 ng·µL^−1^.

For cDNA synthesis, reverse transcription was performed using ReverTra Ace™ Reverse Transcriptase (Toyobo, Osaka, Japan). The RNA was first mixed with RNase-free water to achieve a total volume of 10 µL, which was then denatured at 70 °C for 5 min. The reverse transcription reaction included 1 µL of ReverTra Ace™ enzyme, 5 µL of 5× RT Buffer, 0.75 µL of RNase Ribonuclease Inhibitor, 0.5 µL of a dNTP mixture (2.5 mM each), 2 µL of oligo dT primers (20 mer), and RNase-free water to a final volume of 20 µL. The mixture was incubated at 37 °C for 1 h, followed by enzyme inactivation at 65 °C for 10 min.

Real-time quantitative polymerase chain reaction (RT-qPCR) was conducted using the LightCycler^®^ 480 SYBR^®^ Green I Master (Roche Diagnostics, Mannhein, Germany) and the LightCycler High-Speed/Throughput Real-time PCR System. Each reaction contained 1 µL of cDNA, 10 µL of SYBR Green Master Mix, and 0.5 µL of each gene-specific primer (Appendix A), resulting in a final volume of 20 µL. The PCR program included an initial denaturation at 95 °C for 3 min, followed by 45 cycles of denaturation at 95 °C for 15 s, annealing at 60 °C for 15 s, and elongation at 72 °C for 30 s. The dissociation curve was analyzed after each cycle to confirm the amplification of a single product. Gene expression levels were normalized using the housekeeping gene protein phosphatase 2A (PP2A) and quantified using the 2^−ΔΔC^_T_ method [60]. Each experiment included two biological replicates and two technical replicates.

### 4.9. Quantification of Total Glucosinolate Contents

To quantify total GLS, freeze-dried leaf samples (20 mg) from two-week-old *Arabidopsis* plants treated with CB, CFS, and CS were used. The samples were extracted with 2 mL of aqueous methanol (90% *v*/*v*) by sonication at 40 °C for 1 h. The extracts were then centrifuged at 3500× *g* for 20 min, and the supernatant was collected and filtered. For GLS quantification, 50 µL of the extract was mixed with 150 µL of distilled water and 1.5 mL of 2 mM sodium tetrachloropalladate solution. The mixture was incubated at room temperature for 1 h. The absorbance was measured at 425 nm using a spectrophotometer. A blank sample without the extract was prepared using the same procedure. The total GLS content was calculated using the following formula:$$y=1.40+118.86\times A_{425}$$
where *A*_425_ is the absorbance reading [61].

Each sample measurement was repeated three times, and the data are presented as the mean ± SD.

### 4.10. Statistical Analysis

All data are presented as the mean ± SD unless otherwise stated. Statistical analyses were performed using GraphPad Prism software version 8.4.3 (San Diego, CA, USA). Statistical comparisons between the control and treated samples were conducted using the Student’s *t*-test with significance levels of * *p* < 0.05, ** *p* < 0.01, and *** *p* < 0.001, followed by the two-stage step-up method of Benjamini, Krieger, and Yekutieli.

## 5. Conclusions

In conclusion, this study highlights the potential of recycling CFS as a biostimulant to enhance plant growth and resilience under stress conditions. Specifically, our results demonstrated that CFS treatment significantly increased root length, cotyledon size, rosette size, plant height, and silique formation, indicating enhanced growth in both *Arabidopsis* seedlings and mature plants. Notably, the improvement in drought tolerance of *Arabidopsis* plants treated with CFS is reported for the first time. This increased drought tolerance in CFS-treated plants may be attributed to reduced oxidative stress, increased stomatal closure, and GLS accumulation. This study provides preliminary evidence and new insights into the underlying mechanism of drought tolerance via an auxin signaling-dependent pathway, wherein *IAA5* plays a key role in regulating the downstream transcriptional cascades involved in GLS biosynthesis.

## Figures and Tables

**Figure 1 plants-13-03012-f001:**
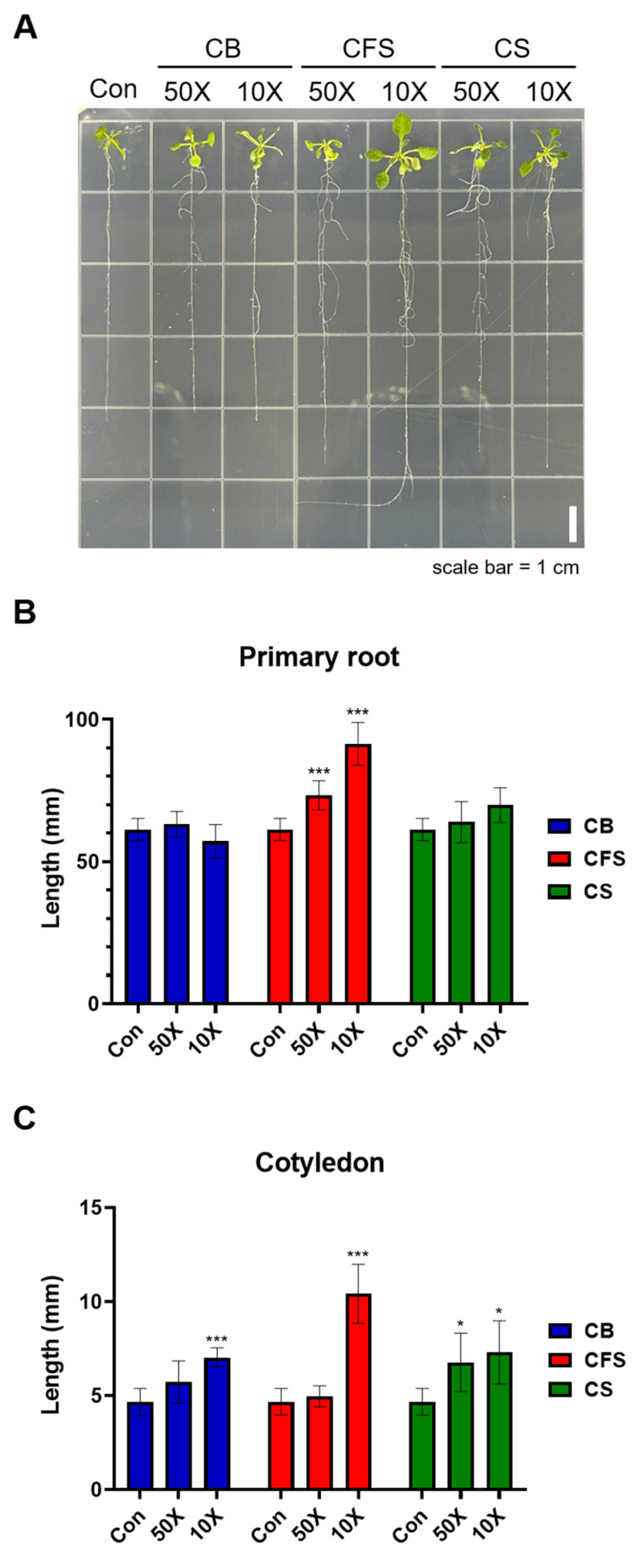
Effects of CB, CFS, and CS supplements on the growth of primary roots and cotyledons of *A. thaliana* seedlings. (**A**) Morphology of 14-day-old seedlings of Col-0 grown with 10% and 2% CB-, CFS-, and CS-supplemented MS. (**B**) Primary root length and (**C**) cotyledon length of *A. thaliana* seedlings. Data represent mean ± SD (*n* = 5). The asterisks indicate significant differences between the control and treatment groups at *** *p* < 0.001 and * *p* < 0.05, determined using the *t*-test. Con, control; CB, *Chlorella* biomass; CFS, *Chlorella* cell-free supernatant; CS, *Chlorella* suspension.

**Figure 2 plants-13-03012-f002:**
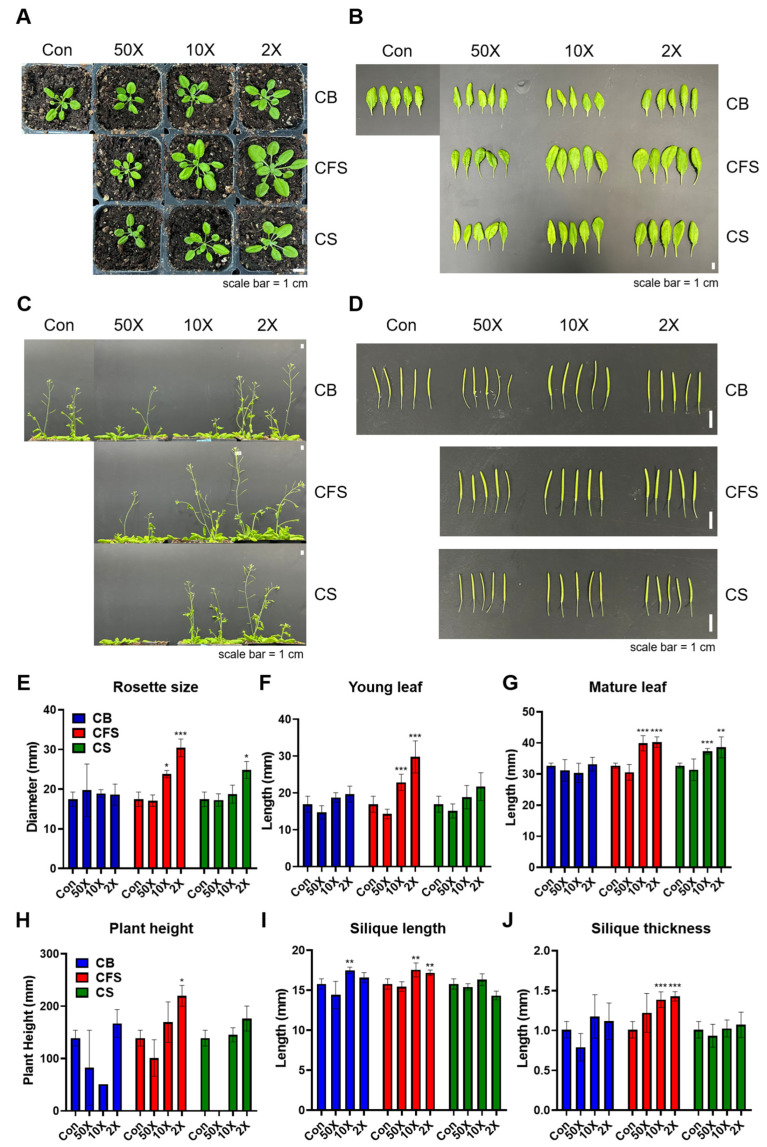
Effects of CB, CFS, and CS treatments on the growth and development of mature *A. thaliana* plants. (**A**) Growth phenotypes of plants grown in soil 2 weeks after transplantation. (**B**) Mature leaf, (**C**) plant height, and (**D**) silique of Col-0 after 4 weeks of treatment with CB, CFS, and CS of *C. vulgaris* diluted 50-, 10-, and 2-fold, respectively. (**E**) Rosette diameter and (**F**) young leaf of “A”. (**G**) Mature leaf of “B”, (**H**) plant height of “C”, (**I**) silique length, and (**J**) silique thickness of “D”. Data represent mean ± SD (*n* = 5). The asterisks indicate significant differences between the treatment and control groups at *** *p* < 0.001, ** *p* < 0.01, and * *p* < 0.05, determined using the *t*-test. Con, control; CB, *Chlorella* biomass; CFS, *Chlorella* cell-free supernatant; CS, *Chlorella* suspension.

**Figure 3 plants-13-03012-f003:**
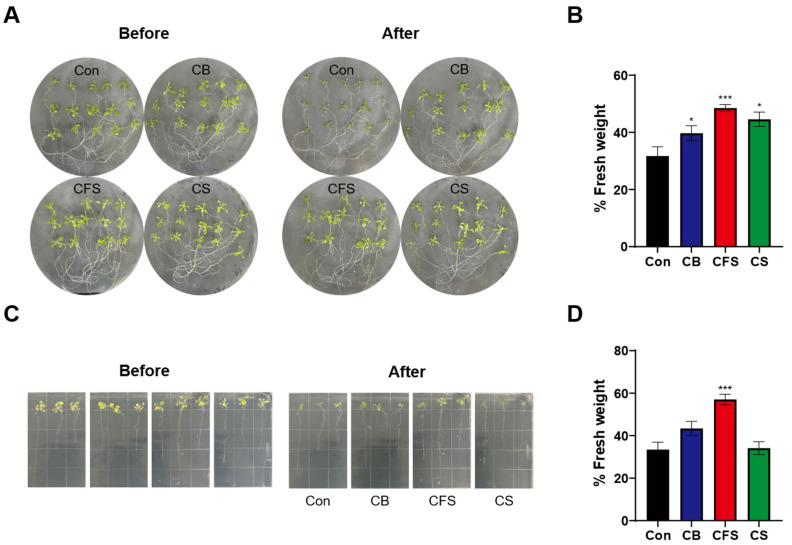
Phenotypes of *A. thaliana* seedlings treated with CB, CFS, and CS under drought stress. (**A**) Phenotypes of 14-day-old seedlings subjected to drought stress by placing on Parafilm and treated with CB, CFS, and CS diluted 10-fold. The seedlings on Parafilm were incubated for 90 min at 22 °C under continuous illumination. (**B**) Fresh weight after drought treatment relative to the initial weight of seedlings. (**C**) Phenotypes of 14-day-old seedlings before drought treatment and after re-watering with CB, CFS, and CS diluted 10-fold for two days. (**D**) Fresh weight after re-watering relative to the initial weight of seedlings. Data represent mean ± SD (*n* = 15). The asterisks indicate significant differences between the treatment and control groups at * *p* < 0.05 and *** *p* < 0.001, determined using the *t*-test. Con, control; CB, *Chlorella* biomass; CFS, *Chlorella* cell-free supernatant; CS, *Chlorella* suspension.

**Figure 4 plants-13-03012-f004:**
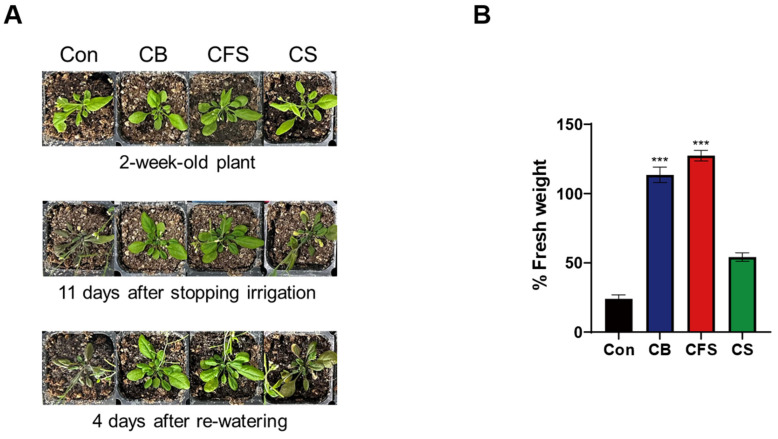
Phenotypes of *A. thaliana* mature plants treated with CB, CFS, and CS under drought stress and after re-watering. (**A**) Phenotypes of two-week-old plants treated with CB, CFS, and CS diluted 10-fold before and after drought stress induced by stopping irrigation for 11 days and after re-watering for four days. (**B**) Fresh weight after re-watering relative to the initial weight of soil plants. Data represent mean ± SD (*n* = 4). The asterisks indicate a significant difference from the control at *** *p* < 0.001, determined using the *t*-test. Con, control; CB, *Chlorella* biomass; CFS, *Chlorella* cell-free supernatant; CS, *Chlorella* suspension.

**Figure 5 plants-13-03012-f005:**
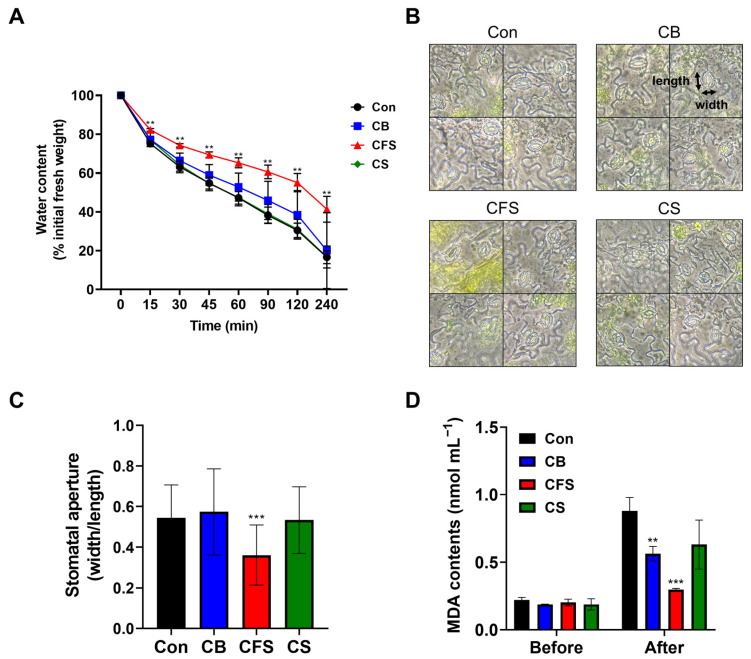
Responses of *A. thaliana* mature plants treated with CB, CFS, and CS under drought stress. (**A**) Assay to determine the water loss rate. Aerial parts of two-week-old well-watered soil-grown plants treated with CB, CFS, or CS were detached and weighed immediately (0 min). They were then allowed to dry naturally under ambient conditions and weighed at the indicated time points. Water content was calculated as the percentage of the fresh weight at 0 min. Data represent mean ± SD (*n* = 3). (**B**,**C**) Stomatal response to drought stress in *A. thaliana* plants treated with CB, CFS, or CS. For stomatal aperture measurement, *n* = 150–200 stomata from six different leaves from three plants grown in the same conditions were observed. (**D**) Effect of drought stress on lipid peroxidation (MDA content) in the leaves of *A. thaliana* treated with CB, CFS, or CS. Data represent the mean ± SD (*n* = 3). The asterisks indicate a significant difference from the control group at ** *p* < 0.01 and *** *p* < 0.001, determined using the *t*-test. Con, control; CB, *Chlorella* biomass; CFS, *Chlorella* cell-free supernatant; CS, *Chlorella* suspension.

**Figure 6 plants-13-03012-f006:**
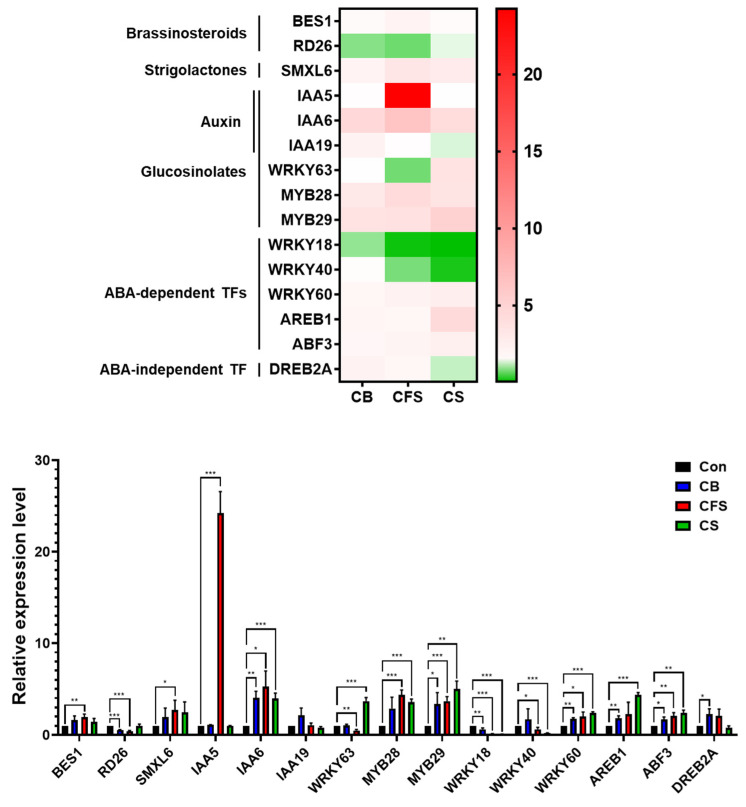
Relative expression levels of drought stress-responsive genes in *A. thaliana* treated with 10× CB, CFS, and CS. Leaves of mature plants grown for four weeks and treated with CB, CFS, and CS were used as samples. Two biological replicates and two technical replicates were performed to quantify target gene expression levels. The asterisks indicate significant differences between the control and treatment groups at *** *p* < 0.001, ** *p* < 0.01, and * *p* < 0.05, determined using the Student’s *t*-test. Con, control; CB, *Chlorella* biomass; CFS, *Chlorella* cell-free supernatant; CS, *Chlorella* suspension.

**Figure 7 plants-13-03012-f007:**
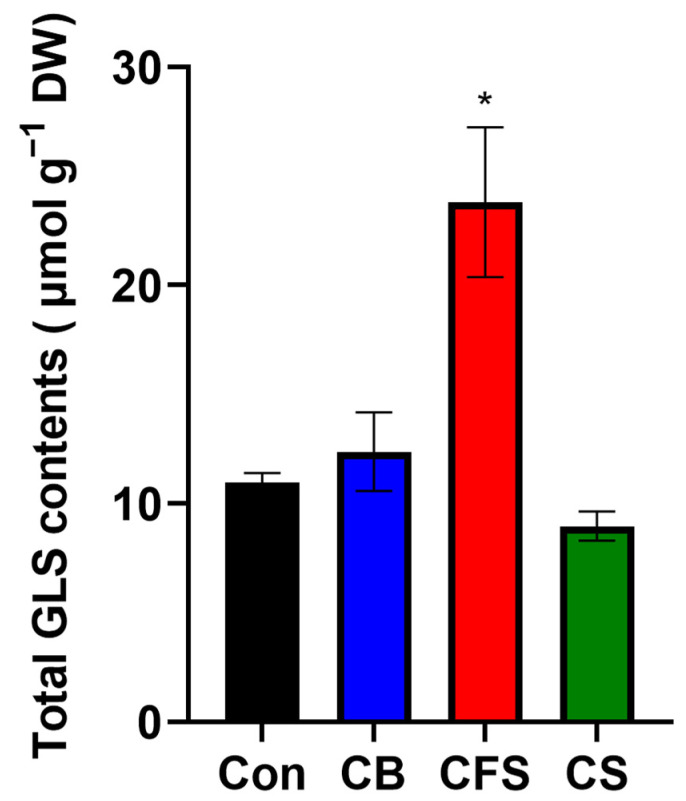
Total glucosinolate content in *A. thaliana* treated with CB, CFS, and CS diluted 10× in water. Leaves of mature plants grown for 4 weeks and treated with CB, CFS, and CS were used as samples. Two biological replicates were performed to quantify the total glucosinolate content. Data represent the mean ± SD (*n* = 3). The asterisk indicates a significant difference between the control and treatment groups at * *p* < 0.05, determined using the *t*-test. Con, control; CB, *Chlorella* biomass; CFS, *Chlorella* cell-free supernatant; CS, *Chlorella* suspension; GLS, glucosinolate.

**Figure 8 plants-13-03012-f008:**
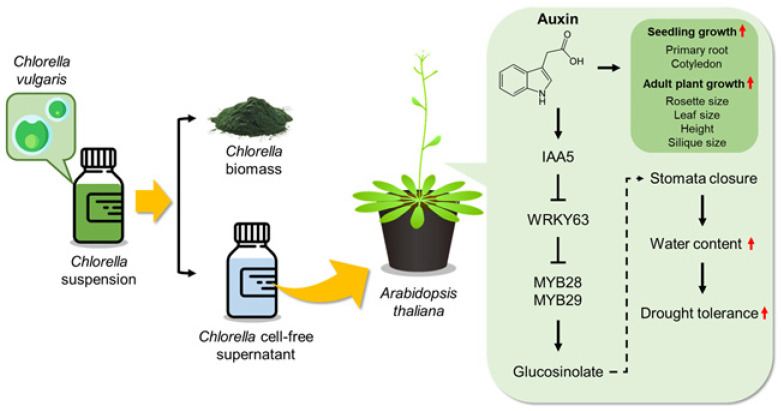
Proposed scheme for the mechanism of enhanced growth and drought tolerance in *A. thaliana* by *Chlorella* cell-free supernatant (CFS). Supplementation with CFS obtained from *Chlorella* suspension improved *Arabidopsis* plant growth and drought resistance. In response to drought stress, CFS induced the expression of *IAA5*, which is responsible for auxin signaling and downstream transcriptional cascades of glucosinolate biosynthesis. Increased glucosinolate production may facilitate stomatal closure and maintain the plant’s water content, improving drought stress tolerance.

**Table 1 plants-13-03012-t001:** *C. vulgaris* treatments used in this study.

Abbreviation	Description
CS	*Chlorella* suspensions (fresh *Chlorella* cultures)
CB	*Chlorella* biomass (centrifuged and resuspended in water)
CFS	*Chlorella* cell-free supernatant (supernatant from centrifuged cultures)

## Data Availability

The raw data supporting the conclusions of this article will be made available by the authors upon request.

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
