# Peer review of "Physiological Effects and Mechanisms of Chlorella vulgaris as a Biostimulant on the Growth and Drought Tolerance of Arabidopsis thaliana"

_plants, 2024, doi:10.3390/plants13213012_

Round 1
Reviewer 1 Report
Comments and Suggestions for Authors
The paper is an investigation into the effects of Chlorella vulgaris as a biostimulant on the growth and drought tolerance of Arabidopsis thaliana
The emphasis on the cell-free supernatant provides new insights into underutilized aspects of microalgal biostimulants, distinguishing it from other studies that focus solely on algal biomass.
The investigation of gene expression changes, particularly the role of auxin signaling and GLS biosynthesis, adds depth to the understanding of how microalgal biostimulants function at a molecular level.
The study includes various concentrations of CFS, CB, and CS treatments, as well as drought stress and recovery phases, offering a robust evaluation of plant responses.
The paper does not explicitly compare the effects of Chlorella vulgaris cell-free supernatant (CFS) or other treatments (biomass, suspension) with the culture medium alone, used for microalgae growth. The focus is primarily on the comparative effects of different fractions of the Chlorella culture (CFS, biomass, and suspension) on Arabidopsis thaliana. The study aims to determine the biostimulant effects of these specific Chlorella products but does not include a control or baseline test with only the microalgae growth medium, which would be useful for distinguishing the effects of the medium's nutrients from those of the algal products themselves; future experiments, including such a comparison would help clarify whether the beneficial effects observed are due to the presence of specific compounds released by Chlorella or are simply a result of the nutrient content in the growth medium.
By the way, the use of Arabidopsis thaliana limits the direct applicability of findings to commercial crops. While Arabidopsis is a model organism, studies on crops like rice or wheat would be necessary to confirm broader agricultural benefits. Although Chlorella vulgaris is compared against its own biomass and suspension, no comparisons are made with other biostimulants, such as seaweed extracts or other microalgae like Spirulina or Dunaliella, or macroalgal commercial produtcs.
However, further research is needed to test these effects in major crops under field conditions. Future studies should focus on validating these findings in economically important crops and exploring the scalability of this approach in diverse agricultural environments.
Comments on the Quality of English LanguageThere are minor grammatical errors scattered throughout the paper. For example, in the phrase, "Chlorella culture suspensions have demonstrated various biostimulant effects on plants" the use of "has" instead of "have" would improve subject-verb agreement.
In the sentence "IAA5 expression increased over 20-fold, IAA6 increased over five-fold, and MYB28 and MYB29 increased over four-fold in CFS-treated plants compared to controls", there is redundancy in the repetitive use of "increased." Rephrasing could enhance readability.
Some sentences are overly long and complex, making it harder to follow. For example, "Results revealed a significant increase in the primary root length for seedlings grown on CFS-treated media compared to those grown on CB- or CS-treated media (Figure 1B, p < 0.001)", could be simplified for better flow.
The study could benefit from more concise and straightforward sentences, particularly in the discussion and results sections, where dense information is provided.
In certain sections, ideas and words are repeated unnecessarily, such as "Arabidopsis plants treated with CFS showed increased growth compared to those treated with CB or CS" being reiterated in multiple forms. This could be trimmed to avoid redundancy.
Some transitions between sections or ideas are abrupt. For instance, the shift from discussing growth effects to drought tolerance could use more smooth transition phrases to maintain logical flow.
The word choice in some sections could be more precise. For example, instead of "increase in the shrinkage ratio", it would be clearer to say "decrease in plant water loss" or "improvement in drought tolerance."
While the paper communicates its scientific ideas effectively, refining the English would improve readability and the professional presentation. A round of proofreading by a native or fluent English-speaking editor, with experience in scientific writing, would enhance the paper's polish and ensure clarity for an international audience.
Author Response
- Major comments
***The paper does not explicitly compare the effects of Chlorella vulgaris cell-free supernatant (CFS) or other treatments (biomass, suspension) with the culture medium alone, used for microalgae growth. The focus is primarily on the comparative effects of different fractions of the Chlorella culture (CFS, biomass, and suspension) on Arabidopsis thaliana. The study aims to determine the biostimulant effects of these specific Chlorella products but does not include a control or baseline test with only the microalgae growth medium, which would be useful for distinguishing the effects of the medium's nutrients from those of the algal products themselves; future experiments, including such a comparison would help clarify whether the beneficial effects observed are due to the presence of specific compounds released by Chlorella or are simply a result of the nutrient content in the growth medium.
Thank you for your comment.
- First, we did not include BG-11 medium treatment in this study for some reasons. vulgaris was grown until stationary phase. At this point, the nutrient composition in culture medium must be different from the initial medium due to the nutrient consumption for growth. In Park’s study, the trace level of nutrients in culture supernatant was identified after exponential phase [1].
- Second, previous studies mainly use Chlorella biomass and its extract, while this study aimed to recycle CFS from Chlorella Therefore, we would like to compare directly to the effect of Chlorella biomass and we did include control group (water only) for normalization.
- Third, we agree with you that the effect might come from the trace nutrient components of BG-11 medium remained in CFS, but this study aimed to directly recycle CFS discarded from Chlorella culture after biomass collection. Therefore, the positive effects from either BG-11’s trace nutrients or biochemical compounds secreted by Chlorella cells will not affect our purpose.
***By the way, the use of Arabidopsis thaliana limits the direct applicability of findings to commercial crops. While Arabidopsis is a model organism, studies on crops like rice or wheat would be necessary to confirm broader agricultural benefits. Although Chlorella vulgaris is compared against its own biomass and suspension, no comparisons are made with other biostimulants, such as seaweed extracts or other microalgae like Spirulina or Dunaliella, or macroalgal commercial produtcs.
Thank you for your contributing opinions. We agree that using other economically important crops could have enhanced the significance of our study as well as the direct applicability of our findings. However, effects of Chlorella on various crops have been reported in previous studies so far, even wheat plants, but very few studies investigated the underlying mechanism [2]. Kindly check lines 283 ~ 289, lines 296 ~ 304 for the previous findings.
Thus, we would like to conduct a systemic study to not only demonstrate the potential of CFS enhancing growth and stress tolerance but also provide new insight into the underlying mechanism regulating stress response. Driven by that purpose, we selected Arabidopsis, known as a robust model for molecular study.
Having said that, we truly appreciate your suggestions, so we clarified this limitation in the discussion part to encourage future study working on other crops based on preliminary evidences in Arabidopsis plant model. Additionally, we understand your concern about lacking positive controls, we added the limitation of study at the end of discussion section to clarify that and encourage future research to study more. Kindly check lines 414 ~ 423.
***However, further research is needed to test these effects in major crops under field conditions. Future studies should focus on validating these findings in economically important crops and exploring the scalability of this approach in diverse agricultural environments.
We truly appreciate your opinion to improve this work. Nevertheless, due to the aim of this study focusing on the responsive mechanism of plants under drought condition behind the effect of Chlorella treatments, the experiments under field conditions might be challenging due to the lack of environmental control. As we agree with you on the further validation of these findings, your suggestion has been introduced in the discussion part, which will be beneficial for future studies to test these effects in major crops under field conditions. Kindly check lines 414 ~~ 423.
- Other comments
***There are minor grammatical errors scattered throughout the paper.
*** There is redundancy in the repetitive use of "increased." Rephrasing could enhance readability.
***Some sentences are overly long and complex, making it harder to follow.
***The study could benefit from more concise and straightforward sentences, particularly in the discussion and results sections, where dense information is provided.
***In certain sections, ideas and words are repeated unnecessarily, such as "Arabidopsis plants treated with CFS showed increased growth compared to those treated with CB or CS" being reiterated in multiple forms. This could be trimmed to avoid redundancy.
***Some transitions between sections or ideas are abrupt.
***The word choice in some sections could be more precise. For example, instead of "increase in the shrinkage ratio", it would be clearer to say "decrease in plant water loss" or "improvement in drought tolerance."
Thank you for your comments. We have revised the manuscript thoroughly, especially important terms and parts as you emphasized.
***While the paper communicates its scientific ideas effectively, refining the English would improve readability and the professional presentation. A round of proofreading by a native or fluent English-speaking editor, with experience in scientific writing, would enhance the paper's polish and ensure clarity for an international audience.
Thank you for your keen comments to improve our manuscript. According to your comments, we revised the whole manuscript for English structure and word choices. Improvements were made to communicate its scientific ideas effectively, especially in results and discussion sections. Lastly, the manuscript underwent one more round of proofreading by an experienced editor in scientific writing before resubmitting to Plants.

Reviewer 2 Report
Comments and Suggestions for Authors
Dear Authors,
I believe the article has a large amount of well-written and well-presented data that could be very useful. However, there are several points that need clear explanation, such as how the controls were conducted, and improving the discussion with support from existing literature.
Majors:
1-L42: “Their lipid con……” Please indicate the reference.
2-Apart from what has been mentioned, I believe the introduction is very well written, covering all the important points, providing a good state of the art, and presenting the hypotheses appropriately
3-L395: “90 min under” I was surprised; isn't 90 minutes a very short time for water stress? Please comment.
4-I haven't found it, and in my opinion, it is essential that the authors clarify what the Control condition (Con) is. I hope the authors have considered that they grew Chlorella in BG-11 medium, and therefore, in the CS and CFS treatments, all the components of that medium are also being added. If they used only water as a control, how do they distinguish the effect of the BG-11 medium? That is to say, in all the CFS and CS treatments, a large amount of BG-11 is being added, especially in the 2X and 10X treatments, where the most noticeable results are observed. Therefore, for their results to be credible, the same amount of BG-11 medium should have been used in each control. Additionally, each condition (CB, CS, and CFS) should have its own control, since each contains a different amount of BG-11. The CB would only have traces, as it is removed by centrifugation.
5-L286: “This difference is likely due to the compounds existing within Chlorella cells being released into the supernatant during separation via centrifugation” I'm sorry, but this explanation is not convincing to me. The authors overlook that they also autoclave, and more importantly, CS must contain all the components found in CFS. So why does CS not have a similar effect to CFS? It could be that the biomass, after autoclaving, is producing some component due to the high temperatures that inhibits the effect observed in CFS. Please discuss this.
6-L296: “Most compounds within microalgae are reportedly not damaged by autoclaving, and it is believed that the destruction of cells results in the secretion of numerous compounds from within the cells into the surrounding medium” Sorry, it seems odd to me. Is there any reference that supports this claim. I believe that most macromolecules are damaged when autoclaved. Please discuss.
7-L299: “This discrepancy may be due to various substances within the cells that can react with compounds outside the cells or a mixture of byproducts released when the cells are destroyed, potentially hindering their proper function as biostimulants” It's a very interesting reflection, but is there bibliography to support it?
8-Since you have seen that IAA is crucial, can Chlorella synthesize IAA? It has recently been observed that Chlamydomonas does so from tryptophan. Could there be tryptophan in the BG-11 medium? Please discuss this possibility.
9-I would recommend that the authors introduce a conclusions section"
Author Response
***L42: “Their lipid con……” Please indicate the reference.
Relevant bibliography was provided. Kindly check lines 45~46.
***Apart from what has been mentioned, I believe the introduction is very well written, covering all the important points, providing a good state of the art, and presenting the hypotheses appropriately
Thank you for your words.
***L395: “90 min under” I was surprised; isn’t 90 minutes a very short time for water stress? Please comment.
At seedling stage, water was halted for 90 minutes to trigger drought stress. The treating period seems short but we believe 90-min treatment properly induced stress in Arabidopsis seedlings according to significant morphological change and water loss, leading to biomass reduction. We added a short statement to clarify this in the discussion section. Kindly check lines 163~166.
***I haven't found it, and in my opinion, it is essential that the authors clarify what the Control condition (Con) is.
The control group was treated with water only and used to normalize the results of other Chlorella treatments. This information was newly added in the text according to your suggestion. Kindly check lines 114~115 and lines 450 ~ 451.
***I hope the authors have considered that they grew Chlorella in BG-11 medium, and therefore, in the CS and CFS treatments, all the components of that medium are also being added. If they used only water as a control, how do they distinguish the effect of the BG-11 medium? That is to say, in all the CFS and CS treatments, a large amount of BG-11 is being added, especially in the 2X and 10X treatments, where the most noticeable results are observed. Therefore, for their results to be credible, the same amount of BG-11 medium should have been used in each control. Additionally, each condition (CB, CS, and CFS) should have its own control, since each contains a different amount of BG-11. The CB would only have traces, as it is removed by centrifugation.
- We did not include BG-11 medium treatment in this study for some reasons. vulgaris was grown until the stationary phase. At this point, the nutrient composition in culture medium must be different from the initial medium due to the nutrient consumption for growth. In line with your opinion, the trace level of nutrients in culture supernatant was identified after exponential phase in Park’s study [1].
- The growth-promoting effect of CFS may be additive effect of a mixture of various compounds derived from trace nutrient of BG-11 medium and live Chlorella cells. It may be challenging to distinguish, and our study aimed to directly recycle the whole CFS fraction. Therefore, the positive effects from either BG-11’s trace nutrients or biochemical compounds secreted by Chlorella cells will not affect our purpose.
- Thank you for your suggestion. Most of previous studies mainly use Chlorella biomass and its extract, so this study aimed to recycle CFS from Chlorella culture. Therefore, we would like to compare directly to the effect of Chlorella biomass. However, we did include control group (water only) for normalization.
***L286: “This difference is likely due to the compounds existing within Chlorella cells being released into the supernatant during separation via centrifugation”. I'm sorry, but this explanation is not convincing to me. The authors overlook that they also autoclave, and more importantly, CS must contain all the components found in CFS. So why does CS not have a similar effect to CFS? It could be that the biomass, after autoclaving, is producing some component due to the high temperatures that inhibits the effect observed in CFS. Please discuss this.
All Chlorella samples were autoclaved before treatment to prevent contamination affecting Arabidopsis cultivation. According to your opinions, we briefly discussed the influence of thermal treatment on nutrient composition of Chlorella culture. Kindly check lines 335 ~351.
***L296: “Most compounds within microalgae are reportedly not damaged by autoclaving, and it is believed that the destruction of cells results in the secretion of numerous compounds from within the cells into the surrounding medium” Sorry, it seems odd to me. Is there any reference that supports this claim. I believe that most macromolecules are damaged when autoclaved. Please discuss.
We have revised this roughly and added a short discussion based on our results and previous findings. According to previous findings, autoclave may destroy the rigid cell wall of Chlorella, leading to the release of various macro- and micro-molecules. Kindly check lines 335~351.
***L299: “This discrepancy may be due to various substances within the cells that can react with compounds outside the cells or a mixture of byproducts released when the cells are destroyed, potentially hindering their proper function as biostimulants” It's a very interesting reflection, but is there bibliography to support it?
Given limited research, we could not find dedicated research proving this hypothesis yet. However, we added a short discussion together with previous questions to make it less ambiguous. We have suggested that this part needs to be studied more in the future. Kindly check lines 335~351.
***Since you have seen that IAA is crucial, can Chlorella synthesize IAA? It has recently been observed that Chlamydomonas does so from tryptophan. Could there be tryptophan in the BG-11 medium? Please discuss this possibility.
BG-11 culture medium was prepared without adding tryptophan or any amino acids, but the possibility of de novo synthesis of phytohormones in microalgal cells has been found in previous studies. Therefore, we added a short discussion about the auxin production in Chlorella culture. Kindly check the discussion section, lines 388 ~397.
*** I would recommend that the authors introduce a conclusions section"
Conclusion was added to summarize our study with highlighted findings and underscore the significance of our study. Kindly check lines 555 ~565.

Reviewer 3 Report
Comments and Suggestions for Authors
In the manuscript entitled “Physiological effects and mechanisms of Chlorella vulgaris as a biostimulant on the growth and drought tolerance of Arabidopsis thaliana”, the authors confirmed the positive effects of biomass and cell-free culture medium of Chlorella vulgaris on the root growth and shoot development of seedlings and Arabidopsis mature plants. The topic is original which can be attractive for many readers. Of note, the manuscript is written well and structured well enough. I believe this manuscript is suitable for publication with the following minor revision;
-Abbreviate the names of Chlorella species after the first introduction in the whole text.
Comments on the Quality of English LanguageThe English style is good.
Author Response
***In the manuscript entitled “Physiological effects and mechanisms of Chlorella vulgaris as a biostimulant on the growth and drought tolerance of Arabidopsis thaliana”, the authors confirmed the positive effects of biomass and cell-free culture medium of Chlorella vulgaris on the root growth and shoot development of seedlings and Arabidopsis mature plants. The topic is original which can be attractive for many readers. Of note, the manuscript is written well and structured well enough.
We truly appreciate your words.
- Minor revision
***Abbreviate the names of Chlorella species after the first introduction in the whole text.
We checked and corrected this in the whole text. Kindly check the revised manuscript.
Round 2
Reviewer 1 Report
Comments and Suggestions for Authors
The revised manuscript now flows better, with clearer transitions between sections. The Introduction provides a stronger contextual background, connecting Chlorella vulgaris as a biostimulant and explaining its potential in agricultural applications.
The sentence structure is now more concise, which improves readability. The complex ideas are broken down more effectively, and redundancies have been minimized.
The experimental results are explained more clearly, with improved figures and statistical significance noted. The use of data to support findings is more explicit, and the comparisons between different treatments (e.g., CFS, CB, CS) are now better explained.
The discussion on gene expression related to drought tolerance (e.g., IAA5, MYB28, MYB29) has been expanded, offering more depth in explaining the molecular mechanisms involved.
While the revised manuscript is much stronger, there are still a few areas for further improvement:
- Appropriate Control Choice (Water vs. BG-11 Medium):
The use of water as a control instead of BG-11 medium raises an important issue regarding the validity of the experimental design. BG-11 medium is nutrient-rich and designed for the growth of microalgae like Chlorella vulgaris, containing essential nutrients that could influence plant growth. By using water as the control, the study may overlook the effects of these nutrients, which could lead to an overestimation of the positive effects attributed to the Chlorella treatments.
Since Chlorella CFS could contain residual nutrients from the BG-11 medium, the observed growth and drought tolerance improvements may be due in part to these nutrients, rather than purely bioactive compounds from Chlorella. A more rigorous control would have been the BG-11 medium itself, as this would help to isolate the effects of the Chlorella products from the medium's nutrients.
The experiment would include both water and BG-11 medium as controls. This dual-control setup would offer a clearer comparison, helping to distinguish between nutrient-driven growth and the specific effects of the Chlorella culture’s biostimulants.
By the way, there are still a few areas for further improvement; comparisons with other Biostimulants: The discussion could benefit from including more comparisons with other commercial biostimulants or species (e.g., Spirulina, Dunaliella), as this would broaden the applicability of the findings. The authors acknowledge that the study is limited by the use of Arabidopsis thaliana as a model organism, and they suggest that future work should focus on economically important crops. Expanding on this aspect in the discussion would be useful.
Comments on the Quality of English LanguageThere is still room for improvement in the grammar and syntax, though the revisions are a significant step up from the earlier version. Some minor grammatical errors persist (e.g., subject-verb agreement and use of tenses), and a final round of proofreading by a native English editor would help polish the manuscript further.
Phrases like "serveing as" instead of "serving as"​ have been corrected, but some awkward wording remains in sections. However, the readability has generally improved.
Author Response
*** The revised manuscript now flows better, with clearer transitions between sections.
*** The Introduction provides a stronger contextual background, connecting Chlorella vulgaris as a biostimulant and explaining its potential in agricultural applications.
*** The sentence structure is now more concise, which improves readability. The complex ideas are broken down more effectively, and redundancies have been minimized.
*** The experimental results are explained more clearly, with improved figures and statistical significance noted. The use of data to support findings is more explicit, and the comparisons between different treatments (e.g., CFS, CB, CS) are now better explained.
*** The discussion on gene expression related to drought tolerance (e.g., IAA5, MYB28, MYB29) has been expanded, offering more depth in explaining the molecular mechanisms involved.
Thank you for your words. We truly appreciate your rigorous revision. Your comments significantly contributed to the improvement of our manuscript.
Major comments
*** Appropriate Control Choice (Water vs. BG-11 Medium):
The use of water as a control instead of BG-11 medium raises an important issue regarding the validity of the experimental design. BG-11 medium is nutrient-rich and designed for the growth of microalgae like Chlorella vulgaris, containing essential nutrients that could influence plant growth. By using water as the control, the study may overlook the effects of these nutrients, which could lead to an overestimation of the positive effects attributed to the Chlorella treatments.
Since Chlorella CFS could contain residual nutrients from the BG-11 medium, the observed growth and drought tolerance improvements may be due in part to these nutrients, rather than purely bioactive compounds from Chlorella. A more rigorous control would have been the BG-11 medium itself, as this would help to isolate the effects of the Chlorella products from the medium's nutrients.
Thank you for your opinions and comprehensive explanation. We agree with your opinions about the benefit of involving dual control, but we don’t think we can re-perform our experiments. Apart from some reasons we mentioned in round 1, we would like to add additional information:
- As we had searched for relevant studies before setting up our experiments, very few studies involved both control (water only) and BG-11 treatment. For example, Ma et al. (2022) investigated biofertilizer potential of microalgae, their results showed no significant difference between control and BG-11 treatment. Likewise, the similar result was obtained in Minaoui et al. (2022) study. Kindly check the cited articles for more information.
- Frankly, this dual-control setup would offer a clearer comparison, helping to distinguish between nutrient-driven growth and the specific effects of the Chlorella culture’s biostimulants. The growth-promoting effect of CFS is likely the additive effect of a mixture of various compounds derived from trace BG-11 components and Chlorella It may be challenging to distinguish that, but it is an interesting topic for future research. Thus, we have briefly discussed this issue that will benefit future research. Kindly check lines 434 ~454.
References:
Ma, C., Cui, H., Ren, C., Yang, J., Liu, Z., Tang, T., ... & Li, R. (2022). The seed primer and biofertilizer performances of living Chlorella pyrenoidosa on Chenopodium quinoa under saline-alkali condition. Journal of Applied Phycology, 34(3), 1621-1634.
Minaoui, F., Hakkoum, Z., Chabili, A., Douma, M., Mouhri, K., & Loudiki, M. (2024). Biostimulant effect of green soil microalgae Chlorella vulgaris suspensions on germination and growth of wheat (Triticum aestivum var. Achtar) and soil fertility. Algal Research, 82, 103655.
*** By the way, there are still a few areas for further improvement; comparisons with other Biostimulants: The discussion could benefit from including more comparisons with other commercial biostimulants or species (e.g., Spirulina, Dunaliella), as this would broaden the applicability of the findings.
Thanks for your suggestion. We have added more relevant information about previous findings of biostimulant effects of other microalgae species, including Spirulina, and compared with our results. Briefly, the results indicate comparable biostimulant effects. These contents have been discussed in our manuscript. Kindly check lines 291 ~312.
*** The authors acknowledge that the study is limited by the use of Arabidopsis thaliana as a model organism, and they suggest that future work should focus on economically important crops. Expanding on this aspect in the discussion would be useful.
We truly appreciate your contributing comments. Previously, we only mentioned this as a limitation of the current study. However, we now add more information to emphasize this aspect, which we hope that will benefit future research. Kindly check lines 328 ~ 335. Additionally, we gave many examples of previous studies on various crops in the discussion.
*** Comments on the Quality of English Language
There is still room for improvement in the grammar and syntax, though the revisions are a significant step up from the earlier version. Some minor grammatical errors persist (e.g., subject-verb agreement and use of tenses), and a final round of proofreading by a native English editor would help polish the manuscript further.
Phrases like "serveing as" instead of "serving as"​ have been corrected, but some awkward wording remains in sections. However, the readability has generally improved.
Thank you for your keen comments, we sent our manuscript to another proofreading process.

Reviewer 2 Report
Comments and Suggestions for Authors
I believe the authors have adequately addressed all of my comments and suggestions, and I accept the paper in its current version.
Author Response
*** I believe the authors have adequately addressed all of my comments and suggestions, and I accept the paper in its current version.
Thank you again for your words. We truly appreciate your comments which helped us significantly improve our manuscript.